# Role of SOX Protein Groups F and H in Lung Cancer Progression

**DOI:** 10.3390/cancers12113235

**Published:** 2020-11-03

**Authors:** Mateusz Olbromski, Marzenna Podhorska-Okołów, Piotr Dzięgiel

**Affiliations:** 1Department of Histology and Embryology, Department of Human Morphology and Embryology, Medical University, 50-368 Wroclaw, Poland; piotr.dziegiel@umed.wroc.pl; 2Department of Ultrastructural Research, Department of Human Morphology and Embryology, Medical University, 50-368 Wroclaw, Poland; marzenna.podhorska-okolow@umed.wroc.pl; 3Department of Physiotherapy, University School of Physical Education, 51-612 Wroclaw, Poland

**Keywords:** non-small cell lung carcinoma, lung squamous-cell carcinoma, lung adenocarcinoma, SOX protein family, SOX7, SOX17, SOX18, SOX30, miRNA, CpG

## Abstract

**Simple Summary:**

The expression of SOX proteins has been demonstrated in many tissues at various stages of embryogenesis, where they play the role of transcription factors. The SOX18 protein (along with SOX7 and SOX17) belongs to the SOXF group and is mainly involved in the development of the cardiovascular system, where its expression was found in the endothelium. SOX18 expression was also demonstrated in neoplastic lines of gastric, pancreatic and colon adenocarcinomas. The prognostic role of SOX30 expression has only been studied in lung adenocarcinomas, where a low expression of this factor in the stromal tumor was associated with a worse prognosis for patients. Because of the complexity of non-small-cell lung cancer (NSCLC) development, the role of the SOX proteins in this malignancy is still not fully understood. Many recently published papers show that SOX family protein members play a crucial role in the progression of NSCLC.

**Abstract:**

The SOX family proteins are proved to play a crucial role in the development of the lymphatic ducts and the cardiovascular system. Moreover, an increased expression level of the SOX18 protein has been found in many malignances, such as melanoma, stomach, pancreatic breast and lung cancers. Another SOX family protein, the SOX30 transcription factor, is responsible for the development of male germ cells. Additionally, recent studies have shown its proapoptotic character in non-small cell lung cancer cells. Our preliminary studies showed a disparity in the amount of mRNA of the *SOX18* gene relative to the amount of protein. This is why our attention has been focused on microRNA (miRNA) molecules, which could regulate the *SOX18* gene transcript level. Recent data point to the fact that, in practically all types of cancer, hundreds of genes exhibit an abnormal methylation, covering around 5–10% of the thousands of CpG islands present in the promoter sequences, which in normal cells should not be methylated from the moment the embryo finishes its development. It has been demonstrated that in non-small-cell lung cancer (NSCLC) cases there is a large heterogeneity of the methylation process. The role of the SOX18 and SOX30 expression in non-small-cell lung cancers (NSCLCs) is not yet fully understood. However, if we take into account previous reports, these proteins may be important factors in the development and progression of these malignancies.

## 1. Summary

It has been proved that the SOX family proteins (SRY-related High Mobility Group (HMG)-box) play a crucial role in the development of the lymphatic ducts and the cardiovascular system. Additionally, it has been recently studied that the SOX18 protein can be involved in wound healing processes and arteriosclerosis disease progression. Moreover, an increased expression level of the SOX18 protein has been found in several malignances, including melanoma, stomach, pancreatic, breast and lung cancers. In breast cancer, SOX18 correlates with the degree of malignancy and the proliferation status. 

Another SOX family protein, the SOX30 transcription factor, is responsible for the development of male germ cells. Additionally, recent studies have shown its proapoptotic character in non-small cell lung cancer cells.

The role of the SOX18 and SOX30 expression in non-small-cell lung cancers (NSCLCs), especially in lung adenocarcinoma (AC) and lung squamous-cell carcinoma (LSCC), is not yet fully understood. However, if we take into account previous reports, these proteins may be important factors in the development and progression of NSCLCs.

Our preliminary studies showed a disparity in the amount of mRNA of the *SOX18* gene relative to the amount of protein. This is why our attention has been focused on microRNA (miRNA) molecules, which could regulate the *SOX18* gene transcript level. So far, we have been able to identify two miRNA molecules that may be involved in this process: miR-7a and miR-24-3p. The fact that their expression level in malignant tumors is altered may be used both in the diagnosis and the treatment of non-small-cell lung carcinomas.

Recent data point to the fact that in practically all types of cancer hundreds of genes exhibit an abnormal methylation, covering around 5–10% of the thousands of CpG islands present in the promoter sequences, which in normal cells should not be methylated from the moment the embryo finishes its development. It has been demonstrated that in NSCLC cases there is a large heterogeneity of the methylation process. Healthy lung cells show a strong hypermethylation of the *SOX18* gene promoter.

Because of the complexity of NSCLC development, the role of the SOX proteins in this malignancy is still not fully understood. Numerous papers published recently show that the SOX family members play a crucial role in the progression of NSCLC.

## 2. Introduction—Lung Cancer Classification

Understanding the molecular mechanism of NSCLCs’ initiation, promotion and progression processes are the main focus of many researchers worldwide. Out of the many cancers identified, lung cancer is the most common in the world (11.6% of all cases) [1]. It mainly occurs in the older population. Lung cancer is by far more common among men (22% of all cancers) than in women (9.5%). Each year, more people suffer and die from NSCLC than from breast, colon and prostate cancers together [1,2,3,4]. This raising trend of NSCLC morbidity has been observed in developing countries [2,3], mainly due to factors such as an aging population, a decrease in physical activity and smoking [1,3].

The main cause of lung cancer is the exposure of the body to harmful, carcinogenic chemical substances [5,6,7]. It is believed that the most common and important risk factor for all histological types of lung cancer, responsible for the development of about 85% of all cases, is smoking [8,9,10,11,12]. The possibility of cancer appearance in smokers increases 20-fold in comparison with non-smokers [9,13]. Furthermore, it was found that both the duration of the smoking habit and the number of daily cigarettes have a directly proportional effect on the appearance of lung cancer [1,4,13,14,15,16,17].

Recently documented studies have led to distinguishing a new type of lung cancer: lung cancer in never-smokers (LCINS) [18]. LCINS is being proved to be caused mostly by exposure to harmful substances, passive smoking and unhealthy eating habits [1,4,10,15,18]. Moreover, LCINS is associated with a range of characteristic mutations different from those present in smokers [19]. In the histological classification, two main groups of lung cancer can be found: small cell lung cancer (SCLC) and non-small-cell lung cancer (NSCLC) [20,21,22]. This distribution has an important clinical significance since these tumors are not only different in terms of their histogeneses, histological structures and disease courses, but also with regard to their methods of treatment and response to the therapies applied [21,22]. NSCLCs are much more common than SCLCs and represent approximately 85% of all lung cancers. Over half of these cancers are detected at an advanced clinical stage—i.e., III or IV [20,21,22,23,24,25,26,27]. NSCLCs are more diverse than SCLCs and are less responsive to common chemotherapy [28]. There are three main histological subtypes of non-small-cell lung cancer: adenocarcinoma (AC), lung squamous-cell carcinoma (LSCC) and large-cell carcinoma (LCC) [21,28].

Adenocarcinoma cases have been reported to be approximately 50% of all NSCLC cases. It differentiates from the alveolar epithelium in the smaller bronchi and the bronchioles [29,30]. Since it is equally common among smokers and non-smokers, it has been postulated that the molecular mechanism of adenocarcinoma progression could be associated with factors other than tobacco smoke. 

Among the different NSCLCs, AC has he most unfavorable prognosis [29]. The second subtype of NSCLC in terms of incidence is LSCC, which constitutes 20–30% of the cases. It originates primarily in the main airways (the larger bronchi) and, because of that, it is often located in the peripheral region. Its pathogenesis is closely linked with an exposure to tobacco smoke. Squamous-cell carcinoma, unlike AC, shows less tendency to metastasize to distant organs. A characteristic histological feature of LSCC is the keratinization of cells and the formation of horny pearls [31,32].

The least frequently occurring type of NSCLC is the LCC (10–15% of all cases) [21,23,28]. This tumor is most often localized in the peripheral segments of the lungs. Histologically, LCC is formed from clusters of large, multilateral cells with a vesicular nucleus and a strongly marked nucleolus [28]. The large-cell cancer group includes tumors poorly differentiated, with a poor prognosis in which the characteristics of LSCCs and ACs have not been identified [33].

Since adenocarcinoma and squamous-cell lung carcinoma are the two most common types of NSCLC, we have focused our attention on these types of tumors. Morbidity is the highest among non-smoking people, women and young people up to 45 years of age. In comparison to other types of NSCLC, AC and LSCC are usually located in a focuse, which enables more efficient and more effective anticancer therapies [2,5,8,34]. 

As many other malignances, lung cancer develops because of an accumulation of genetic mutations and genomic material damage, which lead to tumor transformation [35]. It has been very well documented that in the case of NSCLC progression, most mutations are associated with pathways related to the activation of the epidermal growth factor receptor protein (EGFR)—i.e., RAS/RAF/MEK/ERK and PI3K, as well as AKT/mTOR. Changes in the EGFR pathways lead to an increase in cancer cell proliferation and their ability to metastasize and initiate the angiogenesis process [20,22,26,36]. If one compares normal lung tissue and small-lung cell lung cancer, the EGFR membrane expression is more frequent in NSCLC cases—it has been observed in 40–80% cases of NSCLC, including 24–89% of LSCCs and 23–46% of ACs [37,38,39,40]. 

A better understanding of the molecular mechanisms of NSCLC progression lays in the basis of a new and successful targeted therapy. Therefore, it is important to search for new and more specific therapeutic targets. Bearing this in mind, the SOX protein family members could be a successful anticancer therapy, as well as a new prognostic and progression marker.

## 3. SOX Protein Family

The SOX family genes (SRY-related HMG-box) were firstly investigated and isolated in 1990, mainly because of the fact that they have a common feature: the presence of the High Mobility Group (HMG)-box protein domain, primarily associated with the sex-determining region Y (SRY) [41]. The HMG domain, consisting of 79 amino acids, exhibits about 50% homology in the genes of the SOX group in relation to the SRY gene [42]. All of the SOX proteins act as transcription factors and can be found in different tissues, mostly during embryogenesis, disease processes or carcinogenesis [43]. Recently, the role of SOX proteins in tumor development has been intensively studied, leading to the observation that these transcription factors participate in the pathogenesis of numerous malignant tumors [44]. Up to date, different expression profiles of specific SOX proteins have been reported on different types of cancers, indicating that the same SOX protein can play various roles depending on the type of cancer [45]. For instance, the overexpression of the SOX2 gene in the glioblastoma cell line has been observed despite the fact that the silencing of the SOX2 gene results in a decrease in the invasiveness and migration abilities of the cell line [46]. Additionally, in gastric cancer, a decreased SOX2 expression has also been observed. All this resulted in the induction of cell cycle inhibition and apoptotic processes in said cells [47]. Divergent results have also been obtained in the case of the SOX9 protein. For instance, a reduction in its expression in glioma cells resulted in the inhibition of cell proliferation, while in melanoma the opposite effect (an increased proliferation) could be observed [48,49,50]. The expression levels and mechanisms of SOX2/9 proteins are highly correlated with their role in the cell cycle and the proliferation of cancer cells.

Up until now, 20 proteins have been found to belong to the SOX family. They are divided into eight main groups, from A to H [42,51]. Even though a low-sequence homology among all eight groups has been proven, SOX proteins belonging to the same group show at least 80% homology in the HMG domain [52,53,54]. Proteins belonging to the F and H groups of SOX transcription factors are shown in Figure 1.

SRY (sex-determining region Y), which is the founding member of the SOX protein family, is the only member of the SOXA family group [55]. 

The SOXB group is composed of two subgroups (SOXB1 and SOXB2), which act as transcription factor activators and inhibitors, respectively [42,56]. SOX1, SOX2 and SOX3 proteins that belong to the SOXB1 subgroups show high sequence similarity, hence their identical biological activity in the development of the central nervous system during embryogenesis [56]. The SOXB2 group has two protein members, SOX14 and SOX21, which contain group B homology domain within the SOXB1 group. Additionally, it has been shown that the SOX2 protein can actively bind to the nucleosomes and change the cell fate, which makes SOX2 a pioneer transcription factor [57].

The SOXC group is composed of three members: SOX4, SOX11 and SOX12, which share a well-conserved C-terminal region that forms specific helical conformations, allowing them to show various biological proficiencies [58]. The SOXD group (SOX5, SOX6 and SOX13) was created basically over a conserved domain located in the N-terminal region, which forms two coiled-coil domains, glutamine-rich motif and leucine zipper, allowing them to form stable homo- or heterodimers [59]. SOX8, SOX9 and SOX10 proteins are members of the SOXE group. They contain a distinct dimerization domain located adjacent to the HMG-box and the transactivation domain [60].

Group F contains the proteins SOX7 [61], SOX17 [62] and SOX18 [63], and it has been proven that these play a crucial role in the same pathways as the vascular endothelial growth factors (VEGFs). They are mostly involved in the development of the lymhatic- and cardiovascular systems [64,65]. Since SOX7, SOX17 and SOX18 show high homology in their sequence, they are able to take over each other’s functions under some specific circumstances. It has been very well documented that, in the absence of SOX18 during the development of lymphatic vessels in mice, its role is taken over by SOX7 and SOX17 [66]. 

Group F proteins possess a transactivation domain adjacent to the C-end of the HMG domain, which usually requires interaction with a protein promoter or partner in order to activate transcription [66,67]. For this reason, protein–protein interactions are very important from the functional point of view of this group.

Group G, which in mammals only comprises one protein (SOX15), is predominantly expressed during the embryonic development in the testis, the placenta and the muscular system [68]. 

The SOX30 protein (SRY-box containing gene 30), the only member of group H, has been characterized briefly only in a few species so far. SOX30 has been isolated from mice, humans and *Nile tilapis*, and it has been connected with mammalian spermatogonial differentiation and spermatogenesis [69,70,71,72]. Recently, it has been demonstrated that the SOX30 protein could be a novel epigenetic silenced tumor suppressor that acts through the direct regulation of *p53* in human lung cancer [73,74]. Additionally, there is a whole group of SOX transcription factors in invertebrates which consists of proteins homologous to those of higher organisms [75]. 

There is a more and more experimental evidence of the fact that certain SOX proteins could be used as potential molecular markers for cancer prognosis and become potential therapeutic targets in anticancer therapies in the future. An overview of SOX7, SOX17, SOX18 and SOX30 patterns of expression in different malignancies are shown in Figure 2.

## 4. The Role of SOX Proteins in Lung Tumorigenesis

### 4.1. SOX7

The *SOX7* gene is located in a region of chromosome 8p23.1. Its promotor region shows frequently methylated CpG islands that regulate *SOX7* expression levels [76]. The product of the *SOX7* gene is 388 amino acids in length, with a C-terminal transactivation domain and N-terminal HMG domain [76]. The N-terminal HMG domain contains four phosphorylation sites (S76, S82, S89 and T87) responsible for DNA binding properties. The SOX7 transcription factor shows 70 and 51% sequence similarity to the SOX17 and SOX18 proteins, respectively. It has been shown that many SOX proteins are able to interact with b-catenin, either activating or inhibiting its expression [66]. 

The SOX7 protein is involved in many processes during embryogenesis, such as cardiogenesis and angiogenesis [54,77]. It is also involved in maintaining the balance between proliferation and differentiation through the induction of the expression of the fibroblast growth factor 3 (Fgf-3), VE-cadherin, Gata-4, Gata-6 and laminin-1 genes [78,79,80,81,82]. Additionally, a role in the hematopoiesis process has also been assigned to it. *SOX7* is a gene specifically expressed in Flk+ precursors of the mesoderm [83]. Knockdown of *SOX7* during mesodermal commitment leads to impairment of primitive and definitive hematopoietic and endothelial precursors. Moreover, it has also been proved that SOX7 expression during embryogenesis is sufficient to completely alter the balance between differentiation and proliferation at the initial steps of hematopoiesis [83]. 

The SOX7 protein has been very well described as a tumor suppressor in different types of cancer [81,84,85,86,87,88]. Unlike SOX18 expression, SOX7 expression has been shown to be decreased in lung cancers, which correlates with a poor patient prognosis [89]. There are various epigenetic mechanisms via which gene expression levels can be successfully controlled—e.g., miRNA molecules, promotor methylation, histone modifications, mRNA translation, allele deletion, etc. Up until now, it has been shown that *SOX7* gene expression is being actively controlled by promotor methylation and miRNAs binding to its transcript [85,86,90,91,92]. Hypermethylation of the *SOX7* gene is highly associated with a poor prognosis in myelodysplastic syndrome (MDS) and lung cancer patients [93,94]. 

Recently, it has been proved that miRNA molecules are closely associated with tumorgenesis [95]. Studies performed on hepatocellular carcinomas (HCCs) revealed that *SOX7* regulates the Wnt/β-catenin-signaling pathway via targeting the genes cyclin-1, c-Myc, LEF1 and TCF. miR-184 interacts with the *SOX7* transcript in HCC cells promoting cell proliferation and increases in the sub-G1 phase by regulating cyclin-D and c-Myc [96]. Despite the fact that the molecular mechanisms of miR-184/*SOX7* interaction require further study, it has been proved that the SOX7 protein is essential to cancer cells for increasing their proliferation abilities [77,82,85,90,95].

*SOX7* overexpression has been extensively studied by many researchers. In colon, prostate and NSCLC cancer cell lines, overexpression of the *SOX7* gene leads to the inhibition of proliferation and colony formation processes, as well as apoptosis induction [91,97,98]. It has also been shown that a loss of *SOX7* expression results in acquiring a drug-resistant phenotype against many chemotherapeutic agents by cancer cells. Through the upregulation of genes associated with p38 and MAPK/ERK-BIM-signaling pathway, SOX7 controls the cellular apoptotic mechanism. Additionally, the Food and Drug Administration (FDA)-approved pan-HDAC inhibitor (*Panobinostat*) can elevate SOX7 expression in lung cancer cell lines [99]. 

SOX7 downregulation and the mechanism underlying its reduced expression have also been very well described on breast cancer cell lines and tumors. Similar to what happens in other types of cancer, SOX7 acts as a tumor suppressor in breast cancer pathogenesis. Promoter methylation, as a major mechanism of *SOX7* downregulation, has been identified in six of the nine breast cancer cell lines, although the frequency of methylation was relatively low in comparison to the frequency observed in prostate and colon cancers [85]. This could be due to the fact that breast cancer shows high heterogeneity and *SOX7* methylation occurs only at some specific stages of tumor development. Moreover, the downregulation of SOX7 expression in breast cancers is mediated by additional epigenetic and genetic mechanisms such as histone deacetylation. EZH2 knockdown successfully reduces SOX7 expression [81,85]. 

SOX7 expression patterns have been very well studied in NSCLC cells and tumors. It has been proved that miR-9 is involved in transforming the growth factor beta 1 (TGF-β1)-induced NSCLC cell invasion and adhesion by directly targeting the *SOX7* transcript [100]. miR-9 is upregulated and SOX7 is downregulated in lung cancer cell lines and tissues. Induced overexpression of the SOX7 protein in NSCLC cell lines A549 and HCC827 significantly suppressed cell growth and induced cell apoptosis. [91]. Results show that TGF-β1-induced cell invasion and adhesion are inhibited by miR-9 knockdown, therefore miR-9 promotes NSCLC metastasis and acts as an oncogene in NSCLC [100,101,102,103]. This TGF-β1/miR-9/*SOX7* axis could be a novel therapeutic target in NSCLC. 

There is growing evidence of miRNA molecules that could be involved in NSCLC pathogenesis by targeting the *SOX7* transcript. Yingya et al. showed that miR-24-3p promotes lung cancer cell proliferation, migration and invasion [104]. miR-24-3p promotes tumor growth in xenograft mice by targeting *SOX7*, which suggests that this miRNA molecule could play a role as an oncomiR in lung cancer tumorigenesis by regulating SOX7 expression. 

It has also been found that NSCLC patients with high miR-616 expression levels show significantly shorter overall survival and disease-free survival. The miR-616 molecule regulates the expression of the downstream targets of *SOX7* in lung cancer cells (p-Rb, Cyclin-D1 and c-Myc). This miR-616/*SOX7* downregulation results in an increase in the proliferation and metastasis of NSCLC cells. Similar to miR-24-3p, miR-616 plays an oncogenic role in NSCLC by promoting tumor growth and metastasis. 

Another miRNA molecule proved to have binding properties to the 3’UTR region of the *SOX7* transcript is miR-935. miR-935 has already been reported to downregulate SOX7 expression in liver, gastric and lung cancers [86,105,106]. Downregulation of *SOX7* via miR-935 promotes cell invasion and cell proliferation, as well as cell apoptosis in all the aforementioned types of cancer. This regulation of cell proliferation and invasion is due to the PI3K/Akt-signaling pathway. It has been observed that miR-935 was clearly upregulated in NSCLC tissues compared to non-malignant lung tissue. Knockdown of miR-935 resulting in an increased SOX7 expression leads to the promotion of cell proliferation arrest and cell apoptosis of the lung cancer adenocarcinoma cell line A549 [105]. This could be due to the reduction in the expression level of the Bcl-2 protein and the activity of p-Akt, resulting in an increase in the expression of the proapoptotic Bax protein. 

For decades, scientists have tried to identify the molecular mechanism of cancer cells’ chemoresistance. The majority of anticancer drugs present antiproliferative and proapoptotic properties. Fighting this chemoresistance is one of the challenges that underlie successful anticancer therapy in NSCLC. It has been proved that long non-coding RNAs (lncRNAs) play a crucial role in the drug resistance of various malignances [107]. It is well documented that silencing miR-21-5p expression in lung cancer cell lines enhances their sensitivity to cisplatin (DDP) by inhibiting growth and inducing cell apoptosis. Recent studies have demonstrated that *SOX7* is a direct target of miR-21-5p in NSCLC [90]. MEG3 (Maternally Expressed 3)-imprinted long non-coding RNA gene knockdown in lung cancer leads to enhanced antiproliferative and proangiogenic features of those cancer cells by affecting the *SOX7* transcript. All this results in the inhibition of the chemoresistance properties of NSCLC cells. MEG3 enhances DDP sensitivity of NSCLC cells by regulating the miR-21-5p/*SOX7* axis, causing the A549 cell line to lose its proliferative properties and inducing the cell apoptosis mechanism [108,109,110].

SOX7’s known properties and functions indicate its tumor-suppressor role in multiple kinds of cancers. However, the mechanism underlying SOX7 regulation and its role in NSCLC pathogenesis and chemoresistance still remain poorly understood. Therefore, there is a great need for further investigation of the role of SOX7 in the carcinogenesis mechanism. 

### 4.2. SOX17

It has been proved that the SOX17 transcription factor plays a crucial role in the formation of the endoderm during embryogenesis, giving rise to the pancreas, liver and epithelium of the gastrointestinal and respiratory tracts [111,112,113]. Studies on mice have identified some additional roles of the SOX17 protein, mainly in cardiovascular development, the maintenance of fetal hematopoietic stem cells and angiogenesis [64,114,115]. 

The *SOX17* gene consist of two exons. It is located in the 8q12-q13 region of the human chromosome and was found to encode a 414-amino acid protein with an HMG-box, homologous to SOX7 and SOX18 [116]. Its mRNA has been detected in adult and fetal lungs, the heart, the testis, the spleen, the placenta, the ovaries and the kidneys. Up until now, it has been demonstrated that SOX17 shows almost undetectable expression levels in a variety of cancers cell lines and primary tumors [116,117,118,119,120,121].

Dysregulation of SOX17 is a major key component in the development and progression of numerous types of cancer, including breast and endometria cancer, lung cancer, esophageal and gastric cancer [117,118,122,123,124]. In these types of cancer, a low SOX17 expression has been correlated with a poor patient prognosis. However, SOX17 expression has been reported to increase the sensitivity to cisplatin in endometrial cancer (EC), which explains SOX17 having been proposed as a tumor suppressor [125]. So far, it has been discovered that numerous miRNA molecules play a crucial role in endometrial cancer progression and precipitate in epithelial to mesenchymal transition (EMT). One of these molecules is miR-21-5p, which has also been shown to interact with the *SOX7* transcript [87,90]. The EMT process is crucial for cancer cells to obtain mesenchymal character, allowing them to further metastasize. It has been proved that miR-21-5p’s expression is dysregulated in EC and promotes the progression of EMT in cancer cells [126]. SOX17 protein expression was highest in HEC-1A cells (derived from moderately differentiated EC) than in AN3CA cells (derived from undifferentiated EC), which indicates that SOX17 may be associated with a better cell differentiation. It has been postulated that SOX17 could act as a gene suppressor in the progression of endometrial cancer by directly regulating EMT and cancer cell proliferation. 

It has been proved that the ability to reprogram well-differentiated cells into alternative cell types is associated with the ectopic expression or repression of genes important in embryogenesis processes. The mechanisms via which SOX17 influences progenitor cell behavior in mature respiratory epithelium cells have been identified. The SOX17-dependent progenitor behavior of the respiratory epithelium is associated with both an increased expression of Sca-1 and a variety of genes that promote cell cycle. SOX17 decreases cell cycle inhibitors together with Smad3, resulting in TGF-β1/Smad3-mediated transcriptional response activation [127]. Moreover, SOX17 enhances cyclin D1 expression, decreases TGF-β-responsive cell cycle inhibitors and interacts with Smad3 by blocking Smad3’s DNA binding properties. All this results in SOX17’s role in the regulation of the behavior of multipotent progenitor cells of the lung tissue.

Repair of the endothelial cell barrier during inflammatory injury is crucial for both lung tissue and tissue fluid to maintain homeostasis. SOX17 promotes this endothelial cell regeneration in the endotoxemia model of endothelial injury and acts as a key regulator via the upregulation of cyclin E1-dependent recovery of vascular homeostasis [128]. It has been proved that SOX17 is not only an important transcription factor during embryogenesis and tumorigenesis, but it also plays a crucial role in a mature organism during epithelial cell regeneration, which shows that the SOXF family proteins may have different functions in ECs. The SOX17-HIF-1α-cyclin E1-signaling pathway functions as a central mechanism of EC regeneration. 

Since the majority of SOXF proteins are involved in the same development processes during embryogenesis, it should be no surprise that they share the same role during carcinogenesis. A single missense mutation in a *SOX17* gene can change the DNA-dependent heterodimer formation with Oct4, interfering with the WNT/β-catenin pathway and inhibiting tumor metastasis via Wnt signaling [129,130]. Up until now, it has been proved that SOX2 acts as a tumor propagator in a variety of cancers. Even though SOX17 belongs to the same family as SOX2, it cannot substitute SOX2 in pluripotency reprogramming, but even one single mutation in a *SOX17* sequence can change it to a pluripotency-inducing factor. Cadherin switching is required for increasing cell motility during the EMT process. E-cadherin is converted into N-cadherin, resulting in invasiveness of cancer cells and metastasis [131,132]. It has been proved that the SOX17 N-terminus is required for EC cells to enhance their migration and metastatic properties [130]. SOX17 expression levels are negatively correlated with β-catenin levels. SOX17-HMG-box domain mutations decrease its interaction with the β-catenin gene promoter and inhibit tumor formation. In EC cells, SOX17 binds to the MAML3 promoter, decreasing Wnt pathway protein expression and suppressing cell proliferation. 

*SOX17* has been classified as an oncogene in endometrial cancer for the presence of recurring missense mutations, antagonizing the canonical Wnt/β-catenin pathway [133]. Oct4 protein is crucial for stem cells’ pluripotency and it has been very well documented that it is one of the most important factors during embryogenesis. Moreover, Oct4 also contributes to tumorigenesis in a variety of cancers. It has been shown that Oct4, SOX2 and SOX17 mutants are able to dimerize, forming heterodimers that contribute to tumorigenesis across different cancers [134]. The SOX17-V118M mutant promotes tumorigenic transformation, while wild-type SOX17 cannot support pluripotent reprograming. Therefore, it is of high importance to further investigate cancer-associated mutations and their functional impact on SOX proteins. 

The role of SOX17 proteins in the molecular biology of lung cancer has only been partially investigated. To date, it has been shown that, similar to *SOX7*, *SOX18* and *SOX30*, the *SOX17* promoter region is highly methylated in primary tumors and in plasma samples of NSCLC patients. Moreover, SOX17 promoter methylation in plasma ctDNA significantly influences patients’ survival time. Therefore, the detection of the *SOX17* methylation profile could provide prognostic information about NSCLC patients [135]. 

Similar to other types of cancers, it has also been proved that the SOX17 protein is involved in the Wnt-signaling pathway in NSCLC. It has been reported to promote the degradation of β-catenin/TCF through GSK3 β-independent mechanisms in the Wnt-signaling pathway and it has proved to be an important inhibitor of the canonical Wnt-signaling pathway [135,136,137,138]. Additionally, the epigenetic mechanisms of the SOX17 control have been identified in lung cancer cell lines and primary human lung cancer tissues. The *SOX17* gene is shown to be frequently methylated in NSCLC. SOX17 expression is in turn regulated by promoter region hypermethylation. Moreover, it has been shown that the hypermethylation of the SOX17 promoter region is highly correlated with the female sex. Therefore, *SOX17* methylation status may serve as a potential marker for lung cancer detection and may suggest that females are more vulnerable to SOX17-related lung carcinogenesis [118]. *SOX17* methylation status has also proved to be correlated with lung cancer differentiation. A colony-formation assay shows that SOX17 suppresses lung cancer cell proliferation and inhibits the Wnt-signaling pathway [118].

### 4.3. SOX18

In humans, the SOX18 transcription factor-encoding gene is located in the 20q13.3 locus of the long arm of chromosome 20. It consists of 8901 base pairs and it encodes a 384-amino acid protein [63,139]. It has been reported that the SOX18 protein plays a crucial role during embryogenesis, where it is involved in the development of blood and lymphatic vessels. It has been shown that it takes part in the development of the lymphatic vessels through direct activation of the prospero homeobox-1 (Prox-1) promoter in the fraction of the cells of the venae cavae [65]. Thanks to the expression of the Prox-1 transcription factor, these cells show a lymphatic cell phenotype. They then migrate and combine to form the lymphatic vessels, which in later stages results in the formation of the lymphatic system. In mouse embryos with a silenced SOX18 expression, the processes of formation of the lymphatic vessels are stopped [65].

The SOX18 protein is also involved in the development of the blood vessels and mutations in the *SOX18* gene result in developmental disorders and consequent dysfunctions of the cardiovascular system, leading even to the death of the organism [140]. SOX18 expression has been observed in the epithelial cells (ECs) of developing vessels as well as in emerging follicles in mouse embryos. An impaired development of these structures is present as a result of the mutation of the *SOX18* gene, and it is a characteristic of the presence of the so-called phenotype of ragged (Ra) mice [114,141]. The participation of the SOX18 protein in the development of vascularization has also been confirmed in studies in chicken embryos [142]. Recent studies have shown that some of the mutations in the *SOX18* gene lead to severe developmental disorders—e.g., hypotrichosis-lymphedema-telangiectasia (HLT) syndrome, which is characterized by the presence of lymphedema and the widening of small blood vessels [143].

SOX18 expression has also been shown in different cells and tissues of mature organisms—e.g., in the heart, the lungs, the jejunum, the stomach and the skeletal muscles [63,144]. There is also growing evidence of the role of the SOX18 protein in the tumorgenesis processes [145,146,147,148,149]. *SOX18* gene expression has been observed in a number of human cell lines, such as, for example, in the cells of germ-cell tumors and in gastric, pancreatic and breast cancers [144]. 

SOX18 protein expression has also been observed during the angiogenesis process in mice tumors. Pilot studies carried out by Young et al. have also demonstrated that tumors resulting from the implantation of mouse B16 melanoma cells into mice with inhibited SOX18 expression as well as mice exhibiting a mutated inactive form of SOX18 grew more slowly and characterized themselves by having a lower vascularization density than those tumors with a fully preserved SOX18 function implanted in control mice [150]. Additionally, in vitro studies demonstrated that the SOX18 transcription factor is able to stimulate vein endothelial cells’ (HUVECs’) migration and proliferation, resulting in an increase in the angiogenesis process. Inhibition of SOX18 expression in the breast cancer cell line MCF-7 leads to a decrease in the proliferative and migratory abilities of this cell line due to the destabilization of the actin cytoskeleton [150]. 

During the embryonic development in endothelial cells, SOX18 closely interacts with another transcription factor, the myocyte-specific enhancer factor 2C (MEF2C). In mice with silenced *SOX18* and MEF2C genes, significant dysfunctions of the cardiovascular system take place, which indicates an important interaction between these two proteins in the early stages of angiogenesis [151]. In addition, SOX18 interacts with the matrix metalloproteinase-7 protein (MMP7) in the skin blood vessels. MMP7 degrades the proteins present in the extracellular matrix, which facilitates the so-called blood vessel sprouting in the process of angiogenesis stimulated by SOX18 [152]. It has also been shown that SOX18 interacts with the VCA-1 protein of the endothelial cells [151]. This could be crucial not only in the metastasis processes of the cancer cells, but also in the angiogenesis of the tumor.

An increased expression of all genes from the SOXF group has been shown in tumor tissue compared to unchanged stomach tissue. Additionally, an increased SOX18 protein concentration has been discovered in the blood serum of patients with gastric cancer. In healthy stomach tissue, SOX18 protein expression has occurred in endothelial cells and the muscular layer of the mucosa. When it comes to tumor tissue, SOX18 expression has not been shown in tumor cells but only in stromal cells. Furthermore, cases with a lower SOX18 expression were characterized by a longer recurrence-free survival time. These results suggest that an increase in SOX18 protein expression level may be an adverse prognostic factor in gastric cancer [153]. 

The role of the SOX proteins, including SOX18 and SOX30, in lung cancer is not very well known. Dammann et al. demonstrated a methylation increment of the CpG islands within the *SOX18* gene promoter in 8 out of the 11 tested lung cancer cell lines [154]. Azhikina et al. also studied the difference in methylation profiles of the SOX18 gene between tumor and tumor-surrounding tissues and unchanged lung tissues [155]. These changes in the methylation pattern of the *SOX18* promoter may indicate that early genetic changes could be responsible for morphological and histological phenotypes leading to tumorigenesis. This may have practical clinical implications in the detection of primary tumors and metastatic tumor foci [155]. 

Another feature of cancer cells is the fact that the local hypermethylation of suppressor genes is complemented by a global hypomethylation of the whole genome. Hypomethylation may also apply to specific sequences [156]. The hypomethylation of repeated elements, retrotransposons and introns leads to genomic instability, which may result in an increased predisposition to chromosome rearrangement, deletion or even retrotransposon translocation [157]. Hypomethylation may also lead to the activation of proto-oncogens such as c-Jun, c-Myc and c-Ha-Ras [158]. During cancer progression, an intensification of the phenomenon of global DNA hypomethylation (DNA demethylation) is observed [159]. For instance, a hypomethylation of repeated sequences such as SAT2, LINE1 and ALU in liver cancer can be observed, which correlates with a poor prognosis [160]. 

At least 10% of all the genes that can be expressed are regulated by methylation, and 60% of the genes have local CpG islands. It is for this reason that it can be assumed that in many, if not all, of the signaling pathways, at least one gene is regulated by methylation [161]. If this gene is essential for the regulation of the path, then its silencing can disable the whole signal transmission pathway. If we take into account that 10% of all the 25,000 human genes are regulated by methylation, the are 2500 potential methylation markers [161]. In addition to this, the methylation of the non-coding regions can also have an effect on the expression of genes, so the number of potential markers could be much higher. All of this clearly indicates the need to look for effective malignancy epigenetic markers [162]. 

Much attention is now being given to the identification of the precise region, for which methylation could be a potential diagnostic, therapeutic and prognostic marker, namely the indication of which CpG dinucleotides within a given island are more involved in transcription regulatio, and thus the silencing of which dinucleotides will have the greatest impact on the expression silencing. It has been shown that, in order to silence the transcription, the methylation of the whole CpG island is not necessary, but only the methylation of a few dinucleotides. The study of the CpG islands of many genes has shown that the core regions—i.e., the areas of greatest importance for the transcription regulation—are usually found in a sequence located in the vicinity of the Transcription Start Site (TSS), but can also be seen above or below this location [163]. 

Taking into account previous studies regarding the role of SOX18 in NSCLC, it is very likely that this protein may be an important factor in NSCLC carcinogenesis. Recently published studies have shown the disparity between the mRNA and protein levels of SOX18 in NSCLC, which may be associated with the regulation of the translation by miRNA molecules [149]. miRNAs are a class of short RNA molecules of approximately 20 single-stranded, non-coding nucleotides, whose role is to decrease gene expression at the translation level. 

miRNAs are proved to play many important roles in different biological processes, including cell proliferation and differentiation, apoptosis, embryogenesis and organogenesis. With this in mind, there is no doubt that miRNAs are also crucial factors in tumorigenic processes [164]. miRNAs are not only able to regulate multiple oncogenes and tumor-suppressor gene expression patterns but may also act as an individual oncogene or suppressor. That is why the relationship between the expression pattern of eight different miRNAs and lung adenocarcinoma patients’ survival was studied [165]. 

Patients with a decreased expression level of miR-7a, miR-7b or miR-145, or with an increased expression of miR-17-3p, miR-21, miR-93, miR-106a and miR-155 show a significantly lower survival rate [166,167,168]. Additionally, it has been proved that the inhibition of miRNA expression leads to a reduction in tumor cell proliferation in in vitro studies. 

Olbromski et al. showed that a disparity between the levels of SOX18 mRNA and protein in NSCLC cases is highly associated with the regulation of the translation by miRNA molecules [149]. Two miRNA molecules have been identified that interact with the *SOX18* transcript: miR-7a and miR-24-3p [11,169]. Moreover, this observation has been confirmed in pancreatic ductal adenocarcinoma samples, where the miR-7a molecule targets the *SOX18* transcript, leading to the inhibition of the gp130/JAK2/STAT3-signaling pathway [170].

### 4.4. SOX30

SOX30 (SRY-box containing gene 30) is the only identified member of group H, and it has been characterized merely in a few species so far. It was first isolated from humans, mice and the Nile tilapia, where two transcripts of SOX30 mRNA were found [171,172]. In mice and humans, it is considered to be a crucial transcription factor during spermatogenesis. This protein is highly conserved during evolution, showing more than 76% homology between humans and mice [171]. SOX30 shows the highest similarity in its sequence to group F, suggesting that they may share comparable functions. It has also been shown that changes in methylation are associated with opposite changes in SOX30 expression patterns, where hypermethylation of the SOX30 promoter region might negatively contribute to its expression [173].

It has been discovered that the *SOX30* gene is a tumor suppressor, acting as a transcription factor and binding directly to the p53 promoter. This binding results in the activation of p53 transcription, initiating apoptosis and suppressing tumor formation [73,74]. The study of the literature clearly offers growing evidence of the role of the SOX30 protein in cancer tumorgenesis. SOX30 has already been shown to be downregulated in acute myeloid leukemia (AML), which is highly associated with hypermethylation of the SOX30 promoter region. Overexpression of the SOX30 protein results in a decrease in β-catenin expression leading to Wnt/β-catenin pathway inactivation. Therefore, it is postulated that SOX30 is a crucial transcription factor in AML progression [174]. Additionally, the methylation profile of the SOX30 promoter region has already been identified in chronic myeloid leukemia (CML) patients, but the biological role of this mechanism remains unrevealed. Methylation status correlates with disease progression in CML, but final determination of the impact of SOX30 methylation needs further investigations [175]. 

Recently, Peng et al. investigated the potential prognostic and diagnostic value of the SOX30 protein in breast cancer patients. Similar to other types of cancer, in breast cancer higher SOX30 expression levels correlate with an enhanced disease-free survival (DFS) and an overall survival (OS). This could be due the fact that SOX30 correlates with favorable tumor characteristics (T stage, tumor size, N stage and TNM stage) in breast cancer, or to the promotion of p53 transcription, which results in a decrease in the chemoresistance properties of cancer cells [176]. Since it has been observed that SOX30 is negatively correlated with tumor size but positively correlated with DFS and OS, its further study might contribute to the outcome of breast cancer patients. 

Previous studies clearly showed that the amplification of the chromosomal region 5q31-5q35.3 results in a strongest correlation with OS in advanced ovarian cancer (OC) patients [177]. SOX30 has been located in the chromosome 5q33 within the 5q31-5q35.3 region, which explains why Han et al. have studied whether SOX30 might be considered a potential prognostic marker in OC or not [178]. A statistically significant effect of high SOX30 expression on a better OS in OC advanced-stage patients has been observed. Additionally, a high SOX30 expression favors the response to platin and/or taxol-based chemotherapy. SOX30 acts as a tumor suppressor and inhibits metastasis via an EMT process, which justifies the further study of its biological functions in OC. This may result in the future in the incorporation of SOX30 into the chemotherapeutic treatment of advanced-stage OC patients. 

The potential role of the SOX30 protein in prostate, bladder and colon cancers has also been studied [179,180,181]. In prostate cancer, SOX30 acts as a tumor suppressor, similarly to what happens in other types of cancer. Moreover, it was discovered that a lower SOX30 expression in Prostate Cancer (PC) cells compared to normal tissue cells is caused by the inhibiting effect of miR-653-5p’s interaction with the SOX30 transcript, which results in the downregulation of Wnt/β-catenin signaling. This miR-653-5p/SOX30/Wnt/β-catenin axis may be involved in the progression of prostate cancer [179]. The overexpression of SOX30 reduces Wnt/β-catenin signaling, therefore targeting SOX30 in PC patients could be a promising choice for innovative anticancer therapy. 

Similarly to prostate cancer, SOX30 expression shows a lower expression in bladder cancer (BC) compared to adjacent normal tissue. Moreover, a poor OS and advanced TNM stages are highly associated with a decreased SOX30 expression [180]. The overexpression of SOX30 in bladder cancer cell lines (T24 and 5637) results in the inhibition of cell proliferation and invasion and promotion of apoptosis mechanisms. Therefore, the overexpression of SOX30 in BC cells could inhibit the progression and development of bladder cancer.

Guo et al.’s experimental studies showed interesting results on the role of miRNA-645 as an oncogenic regulator in colon cancer (CC). It was revealed that SOX30 is one of the key targets of miR-645 in CC [181]. Differentially expressed miRNAs in colon samples compared to paired non-cancerous mucosa were investigated, out of which only one miR-645 was shown to be upregulated in CC cells acting as an oncogenic regulator, promoting proliferation and resistance to apoptosis. miR-645 presents a negative association with SOX30 levels. However, there are other mechanisms apart from the inhibition of SOX30 that also contribute to the oncogenic effect of miR-465 in CC. 

Through a methylation-sensitive representational difference analysis, it has been demonstrated that the *SOX30* gene promoter is also highly methylated in human lung cancers. It has also been proved that the *SOX30* gene is frequently silenced or downregulated in lung cancer cell lines and lung cancer samples but expressed in normal human lung tissue. This effect could be related to its hypermethylation [73,74]. Data published by Han et al. showed different methylation patterns of the SOX30 promoter region in lung cancer, peri-tumoral lung tissue and normal unchanged lung tissues. SOX30 methylation may be a putative epigenetic biomarker for lung cancer, and the knockdown of SOX30 inhibits cell apoptosis and proliferation. Moreover, SOX30 significantly inhibits tumor growth in nude mice, classifying SOX30 as a tumor suppressor in lung carcinogenesis. The upregulation of p53 and its downstream targets by SOX30 explain these antiapoptotic and antiproliferative features. With the use of the p53 inhibitor pifithrin-α, the authors were able to demonstrate the diminished effect of SOX30 overexpression on cell proliferation and cell death, by direct binding to the promoter region of p53 and other downstream factors (BAX, PMAIP1 and p21). 

A different study by Han et al. shows that a high expression of SOX30 is associated with a favorable OS in lung adenocarcinoma patients [73]. SOX30 expression correlates with both the histological type and clinical stage of NSCLC, and this transcription factor may be considered as an independent prognostic factor for OS in AC. It is postulated that different expression levels of SOX30 in lung AC and LSCC samples explain their different prognostic values among these types of NSCLC. In order to clarify this further, the effect of SOX30 on cell proliferation and apoptosis was examined in lung adenocarcinoma and squamous-cell carcinoma cell lines by using the gain-of-function mice model. The experiments showed that SOX30 is a tumor suppressor in AC but has no effect on cell proliferation, cell cycle or apoptosis in LSCC, which underlies the differences in survival outcomes between these two types of NSCLC. SOX30 could be an independent prognostic marker for increased OS of AC patients and those who are at clinical stage II, with positive lymph nodes, at a histological grade of 2 or 3 [73]. 

Cell junction molecules such as desmosomes play a crucial role in inhibiting tumor growth and motility via metastasis processes [182,183,184]. Desmosomal proteins are shown to be deregulated in a variety of cancers, including lung cancer [185]. Moreover, they can serve as independent prognostic and diagnostic markers, such as is the case, for instance, with plakophilin 1 (PKP1) in LSCC [186,187], PKP3 and its tumor growth promotion in lung cancer [188], and the decreased levels of desmocolin 1 (DSC1) in NSCLC [189], which are shown to be associated with a poor prognosis. It has been proved that SOX30 expression can be used as an independent prognostic factor for AC patients but not for LSCC ones, and its proapoptotic and antiproliferative functions play a crucial role only in AC tumorigenesis [73,74]. Many studies showed that intercellular junctions, specially desmosomal genes, can be regulated via different transcription factor interactions with their promoter regions. The differences in desmosomal gene expressions may lie at the base of the differences in regulatory patterns of tumorgenesis between the AC and LSCC subtypes. In AC patients, SOX30 suppresses both Wnt and ERK signals in a desmosomal gene-dependent pathway. Taking all of this into consideration, it could be suggested that the SOX30-desmosomal gene axis may be involved in the development of lung tumors. 

Similarly to other SOX proteins, it has also been proved that SOX30 is involved in regulation in the Wnt-signaling pathway by interacting with the *CTNNB1* gene (β-catenin) in AC and other types of cancer [190,191,192,193] but not in LSCC cases. SOX30 is shown to play a different role in tumor metastasis, and it regulates the Wnt/β-catenin-signaling pathway differently in AC and LSCC patients. In early stages of AC, increased SOX30 levels repress Wnt/β-catenin signaling by direct binding to the CTNNB1 promoter region, resulting in an unfavorable prognosis of AC patients only [190,194]. The reason why SOX30 is unable to bind to the CTNNB1 promoter region in LSCC patients is still unknown. Two causes could be (1) the CTNNB1 promoter being occupied by others factors or (2) some crucial mutations taking place in the promoter region.

In order to investigate the role of the SOX30 transcription factor in metastasis and the Wnt/β-catenin-signaling pathway, Han et al. constructed an induced a SOX30 knockout C57BL/6 mouse model [195]. Experimental data analysis clearly indicates that SOX30 expression in mice strongly reduces the metastatic potential of B16 cells to lung tissue. This could be due to the competition of SOX30 with TCF in a similar region of β-catenin or caused by the regulation of β-catenin stability. Previous studies showed that some SOX proteins are able to actively bind to β-catenin in the region that overlaps the TCF binding region [99,192,193,196,197]. The N-terminus of SOX30 is proved to be required for interacting with β-catenin proteins, and it competes with TCF4 in a similar region of β-catenin. Negative correlations between SOX30 and β-catenin were found in pairs of lung cancerous and peri-cancerous tissues, as well as in the nuclei and cytoplasm of these clinical pairs. Patients with a positive SOX30 nuclear expression showed longer overall survival, which indicates that this protein is a key regulator of the metastasis in lung cancer. Therefore, considering the SOX30 protein as a new potential target in novel anticancer therapy in NSCLC seems to be more than justified.

## 5. Conclusions

Out of all lung cancer types, AC and LSCC are the two most common types. It is believed that the most common and important risk factor for all histological types of lung cancer, responsible for the development of about 85% of all cases, is smoking. The main disadvantage of using standard chemotherapy is the low selectivity in relation to cancer cells, because the substances used in the treatment of lung cancer are also harmful for healthy, unchanged body cells. The data presented in this manuscript show that SOX7, SOX17, SOX18 and SOX30 are proteins involved in the progression of NSCLC. Divergent mRNA and SOX protein levels in NSCLC and normal lung tissue have drawn the attention of scientists to the potential role of miRNA molecules interacting with the transcript of the *SOX18* gene, and on the possibility of *SOX18* and *SOX30* as methylation promoters. This may be crucial in the development of effective anticancer-targeted therapy. These epigenetic modulations show how complex and complicated the molecular pathogenesis of the development of AC and LSCC is. Figure 3 presents SOX18 and SOX30 immunochemical reactions and shows their expression pattern in NSCLC cases.

A better comprehension of the molecular mechanisms of the progression of malignant tumors may lead to the discovery of modern and highly specific antitumor therapies. Therefore, more effort has to be made in trying to discover new and more effective therapy schemes based on markers that could be potential targets. 

The prospect of the application of miRNA which interacts with the SOX genes’ transcripts in the treatment of NSCLC seems to be promising, for there are already many other similar miRNA molecules with proven anticancer functions. The variable levels of the miRNA molecules as well as the DNA methylation profile of the SOX18 and SOX30 promoters can be detected in the patient’s blood, which additionally increases the availability and universality of their use in the diagnosis of lung cancer, as they provide information not only on the type of tumor, but also on the development stage. In this sense, the SOX protein family appears to be an auspicious element in anticancer therapy.

## Figures and Tables

**Figure 1 cancers-12-03235-f001:**
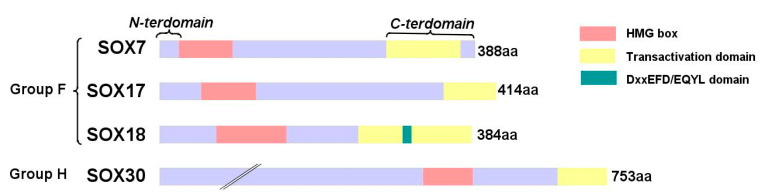
Classification and structure of SOXF and SOXH family groups with indication of C-terminus, N-terminus, High Mobility Group (HMG)-box, transactivation and DxxEFD/EQYL domains and proteins length.

**Figure 2 cancers-12-03235-f002:**
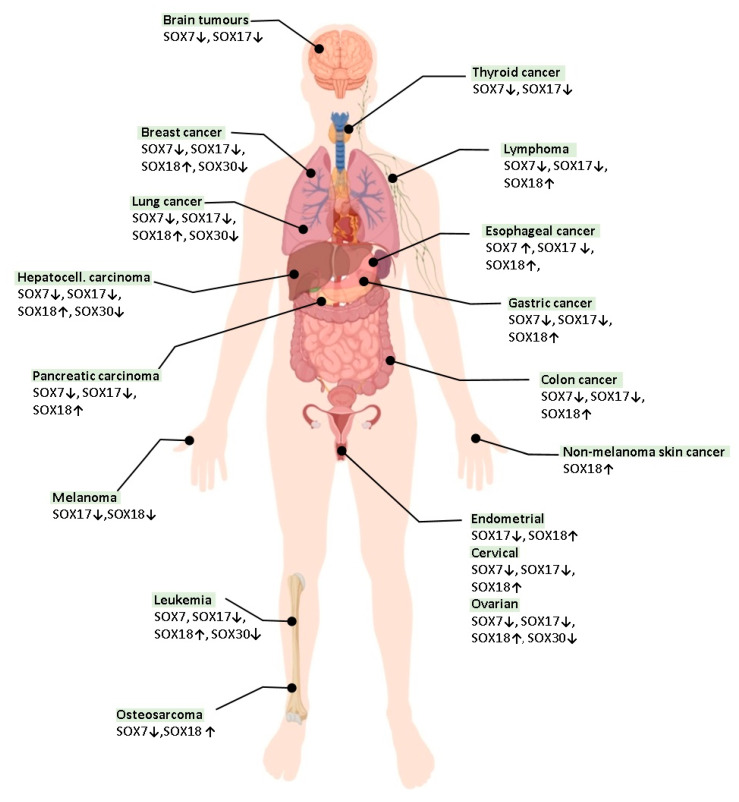
Overview of SOX protein expression patterns in different types of human malignances. Arrows indicate decreased (↓) or increased (↑) expression levels of SOX7, SOX17, SOX18 or SOX30. Parts of the figure were drawn and downloaded from bioRender (https://biorender.com/).

**Figure 3 cancers-12-03235-f003:**
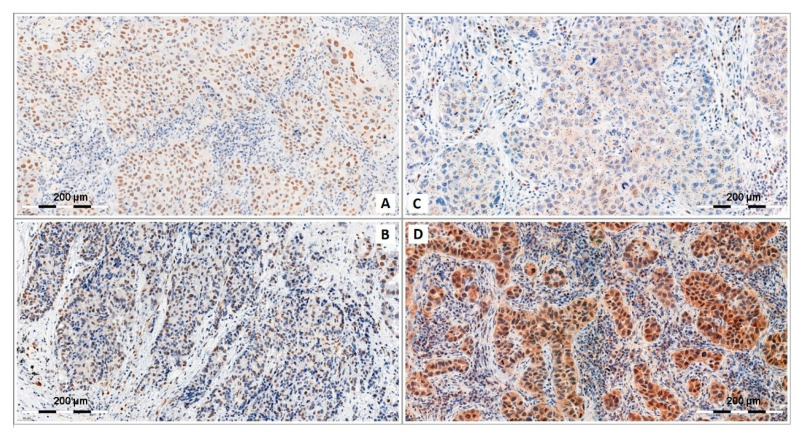
Different expression of nuclear SOX18 (**A**,**B**) and nuclear and cytoplasmic SOX30 (**C**,**D**) proteins in non-small cell lung cancers (NSCLCs). Original magnification, ×600.

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
