# Peer review of "Role of SOX Protein Groups F and H in Lung Cancer Progression"

_cancers, 2020, doi:10.3390/cancers12113235_

Round 1
Reviewer 1 Report
Summary
This is a nice review of a little-studied group of the SOX family, and their role in cancer. This review could benefit from a bit more direct speculation of the role of these proteins in lung cancer, specifically NSCLC. A few minor points are attached.
Major Points
- None noted
Minor Points
- SOX family is general in many neural cells and epithelial cell types, not just ducts and cardiovascular systems (Arnold, Cell Stem Cell, 2011)
- Line 131 – Do the authors mean to reference NSCLC, rather than SCLC? Otherwise this statement is a bit confusing and the point would need clarification.
- Line 144 – SOX2 family members are also expressed in many adult stem cell populations (neural stem cells, neuroendocrine progenitors, etc)
- Lines 149-156 – Some discussion of SOX2 in different cell types may be warranted. Effects of SOX2/9 misregulation can be varied if targeted to tumor propagating cell rather than the tumor parenchyma.
- Lines 163-168 – It is an oversimplification to describe SOX2 as strictly a transcriptional co-activator, as it has been assigned both activation and repressive roles depending on co-factors recruited (Boyer, Cell, 2005, amongst others)
- Section 3, SOX2 protein family – This section could benefit from a discussion of the SOX family as potential pioneer factors (Soufi, Cell, 2015).
- Do miRs 21-5p, 24-3p, 616 and 935 share seed sequences or are these independently targeted miRs to SOX7?
- Line 280 – Scientist[s]
- Line 305 – What cell type is SOX17 found in the lung? Does this shed light on the pathogenesis of NSCLC?
- Is SOX2 18 observed the NSCLC stroma, to perhaps enhance vascularization?
- With all the discussion of promoter methylation, why has DNA demethylation therapies (5azaC, 5azadC) not yielded major results in lung cancer?
- Line 476 – explanation of miRNAs is confusing as miRs were discussed extensively earlier (SOX7 section).
- Are SOX7, SOX17, SOX18 and SOX30 uniquely overexpressed in lung cancers? Is there something unique about these members of the SOX family, or is there a genral misregulation of the entire SOX family in lung cancer?
Author Response
- SOX family is general in many neural cells and epithelial cell types, not just ducts and cardiovascular systems (Arnold, Cell Stem Cell, 2011)
That’s true, but it has been proved that SOXF group family members play a crucial role mostly in the cardiovascular system development during embryogenesis.
- Line 131 – Do the authors mean to reference NSCLC, rather than SCLC? Otherwise this statement is a bit confusing and the point would need clarification.
Indeed, the Reviewer found an error which has been corrected in the text.
- Line 144 – SOX2 family members are also expressed in many adult stem cell populations (neural stem cells, neuroendocrine progenitors, etc)
This line refers to all of the SOX members, including SOX2, but without any division into specific groups.
- Lines 149-156 – Some discussion of SOX2 in different cell types may be warranted. Effects of SOX2/9 misregulation can be varied if targeted to tumor propagating cell rather than the tumor parenchyma.
A short description has been added to the text: “The expression levels and mechanisms of SOX2/9 proteins are highly correlated with their role in the cell cycle and the proliferation of cancer cells”.
- Lines 163-168 – It is an oversimplification to describe SOX2 as strictly a transcriptional co-activator, as it has been assigned both activation and repressive roles depending on co-factors recruited (Boyer, Cell, 2005, amongst others)
It has been pointed out in the text that the SOX2 protein can act as an activator or an inhibitor [42,56].
- Section 3, SOX2 protein family – This section could benefit from a discussion of the SOX family as potential pioneer factors (Soufi, Cell, 2015).
Since this paper is mostly focused on the groups SOXF and SOXH, we didn’t add extra details about other SOX protein family members. A short description to this observation has been added to the manuscript. [line 168] “It has been shown that the SOX2 protein can actively bind to the nucleosomes and change the cell fate, which makes SOX2 a pioneer transcription factor”.
- Do miRs 21-5p, 24-3p, 616 and 935 share seed sequences or are these independently targeted miRs to SOX7?
They are independent factors.
- Line 280 – Scientist[s]
It has been corrected in the paper.
- Line 305 – What cell type is SOX17 found in the lung? Does this shed light on the pathogenesis of NSCLC?
In this line we make reference to all the data regarding SOX17 expression patterns in different cancer cell lines and tumors.
- Is SOX2 18 observed the NSCLC stroma, to perhaps enhance vascularization?
The differences between SOX18 expression patterns in NSCLC and NMLT have been described in lines 490-495. These differences are caused due to miRNAs posttranscriptional interaction with the SOX18 transcript.
- With all the discussion of promoter methylation, why has DNA demethylation therapies (5azaC, 5azadC) not yielded major results in lung cancer?
Unfortunately, we do not have sufficient knowledge to answer this question, because this is not our field of expertise. We just simply made some reference to other scientists’ observations.
- Line 476 – explanation of miRNAs is confusing as miRs were discussed extensively earlier (SOX7 section).
This sentence has been deleted from the manuscript.
- Are SOX7, SOX17, SOX18 and SOX30 uniquely overexpressed in lung cancers? Is there something unique about these members of the SOX family, or is there a general mis regulation of the entire SOX family in lung cancer?
SOX7, SOX17, SOX18 and SOX30 proteins show different expression patterns among NSCLC cases. It has been proven that SOX7, SOX17 and SOX30 are decreased, while SOX18 is increased in NSCLC cases. Additionally, the correlation between SOX18’s mRNA and protein levels is unique in NSCLC in comparison to other of cancer. For this reason, miRNAs regulation by the SOX18 transcript was also investigated. All of this has been also mentioned in the manuscript.
Reviewer 2 Report
SOX family proteins regulate self-renewal and differentiation in stem/progenitor cells of specific tissues, and their abnormal expression contributes to tumor progression. In this review, the authors summarize the findings regarding group F genes SOX7, SOX17, and SOX18 as well as group H gene SOX30. The authors also focus on the crosstalk between miRNAs and these SOX proteins. Comments are listed as follows:
1. Figure 2 is hard to be interpreted. In lung cancer, for example, SOX17 and SOX18 are amplified whereas SOX7 is deleted in lung tumors from LUNG-TCGA database. However, figure 2 shows decreased expression of SOX7, SOX17, SOX18, and SOX30 in lung cancer. The sources of data are unclear.
2. Figure legends are unclear for figure 3. What are the important messages conveyed from this figure?
3. It is too risky to claim the SOX17 is a tumor suppressor since it is amplified in cancers from lung and endometrium and displays a hotspot mutation with the driver mutation feature in some of endometrial cancer.
4. The description of miRNA in the section of SOX18 (line 479-489) is not relevant.
5. Following sentences are misleading:
a. “it has been proved that the SOX7 protein is essential to cancer cells for increasing their proliferation abilities (line 231-232).”
b. “SOX17 decreases cell cycle inhibitors for Smad3, resulting in TGF-β1/Smad3mediated transcriptional response activation [126] (line 329-330). “
c. “The upregulation of p53 and its downstream targets by SOX30 explain this antiapoptotic and antiproliferative features (line 566-567)”.
d. “Divergent mRNA and SOX protein levels in NSCLC and normal lung tissue have drawn the attention of scientists to the potential role of miRNA molecules interacting with the transcript of the SOX18 gene, and on the possibility of SOX18 and SOX30 as methylation promoters (line 626-628).”
Author Response
- Figure 2 is hard to be interpreted. In lung cancer, for example, SOX17 and SOX18 are amplified whereas SOX7 is deleted in lung tumors from LUNG-TCGA database. However, figure 2 shows decreased expression of SOX7, SOX17, SOX18, and SOX30 in lung cancer. The sources of data are unclear.
The figure was created basically on the data collected and described in this paper. Indeed, scientific data show a decreased expression levels of SXO7, SOX17 and SOX30 in NSCLC, but the SOX18 shows an increased expression level, which also shown in figure 2.
- Figure legends are unclear for figure 3. What are the important messages conveyed from this figure?
The figure presents typical IHC images of SOX18 and SOX30 expression patterns in NSCLC cases. The Figure reference has been also placed in the manuscript.
- It is too risky to claim the SOX17 is a tumor suppressor since it is amplified in cancers from lung and endometrium and displays a hotspot mutation with the driver mutation feature in some of endometrial cancer.
This sentence could be a simplification, but was mostly built on the base of many publications.
- The description of miRNA in the section of SOX18 (line 479-489) is not relevant.
Those lines are just a short introduction for readers who need some introduction to the miRNAs field.
- Following sentences are misleading:
- “it has been proved that the SOX7 protein is essential to cancer cells for increasing their proliferation abilities (line 231-232).”
The references to this conclusion have been added to the text. [77,82,85,90,95]
- “SOX17 decreases cell cycle inhibitors for Smad3, resulting in TGF-β1/Smad3mediated transcriptional response activation [126] (line 329-330). “
“For” has been changed to “together with”, which may add some more clarity to the sentence.
- “The upregulation of p53 and its downstream targets by SOX30 explain this antiapoptotic and antiproliferative features (line 566-567)”.
It is simply due to the fact that p53 has been proven to play a crucial role in cell apoptosis and proliferation processes.
- “Divergent mRNA and SOX protein levels in NSCLC and normal lung tissue have drawn the attention of scientists to the potential role of miRNA molecules interacting with the transcript of the SOX18 gene, and on the possibility of SOX18 and SOX30 as methylation promoters (line 626-628).”
Despite the high mRNA expression levels of Sox18, we were not able to observe the same increased protein expression patterns. We wanted to investigate whether miRNA molecules could be involved in the posttranscriptional regulation of the SOx18 transcript.